# Effects of Intravitreal Ranibizumab Injection on Peripheral Retinal Microcirculation and Cytokines in Branch Retinal Vein Occlusion with Macular Edema

**DOI:** 10.3390/medicina59061053

**Published:** 2023-05-30

**Authors:** Kanako Yasuda, Hidetaka Noma, Tatsuya Mimura, Ryota Nonaka, Shotaro Sasaki, Noboru Suganuma, Masahiko Shimura

**Affiliations:** 1Department of Ophthalmology, Hachioji Medical Center, Tokyo Medical University, 1163, Tatemachi, Hachioji, Tokyo 193-0998, Japan; kana6723@yahoo.co.jp (K.Y.); nort310317@gmail.com (R.N.); hanshinchelsea@yahoo.co.jp (S.S.); h-ort@tokyo-med.ac.jp (N.S.); masahiko@v101.vaio.ne.jp (M.S.); 2Department of Ophthalmology, Teikyo University School of Medicine, 2-11-1 Kaga, Itabashi-ku, Tokyo 173-8606, Japan; mimurat@med.teikyo-u.ac.jp

**Keywords:** relative flow volume, vessel width, laser speckle flowgraphy, macular edema, branch retinal vein occlusion, monocyte chemotactic protein-1, interleukin-8, platelet-derived growth factor

## Abstract

*Background and Objectives*: To investigate peripheral blood flow in retinal vessels and vessel diameters after intravitreal ranibizumab injection (IRI) and the relationship between these parameters and cytokines in branch retinal vein occlusion (BRVO) with macular edema. *Materials and Methods*: We assessed relative flow volume (RFV) and the width of the main and branch retinal arteries and veins in the occluded and non-occluded regions before and after IRI in 37 patients with BRVO and macular edema. Measurements were made using laser speckle flowgraphy (LSFG). When performing IRI, we obtained samples of aqueous humor and analyzed them using the suspension array method to evaluate vascular endothelial growth factor (VEGF), placental growth factor (PlGF), platelet-derived growth factor (PDGF)-AA, soluble intercellular adhesion molecule (sICAM)-1, monocyte chemoattractant protein 1 (MCP-1), interleukin (IL)-6, IL-8, and interferon-inducible 10-kDa protein (IP-10). *Results*: In both retinal regions, before and after IRI, the RFV in the main artery and vein showed a significant correlation with the summed RFV in the respective branch vessels 1 and 2. In the occluded region, the RFV in the main vein was significantly negatively correlated with MCP-1, PDGF-AA, IL-6, and IL-8; the RFV in branch vein 1 was significantly negatively correlated with PlGF, MCP-1, IL-6, and IL-8; PDGF-AA was significantly negatively correlated with the width of the main and branch veins; and the RFVs of the main artery and vein decreased significantly from before to 1 month after IRI. *Conclusions*: Contrary to expectations, the study found that anti-VEGF therapy does not affect RFV in arteries and veins in patients with BRVO and macular edema. Furthermore, retinal blood flow is poor in patients with high MCP-1, IL-6, and IL-8. Finally, high PDGF-AA may result in smaller venous diameters and reduced retinal blood flow.

## 1. Introduction

The most common retinal manifestation of lifestyle diseases such as arterial sclerosis and hypertension is diabetic retinopathy, followed by branch retinal vein occlusion (BRVO) [1,2]. Patients with BRVO can become visually impaired, primarily due to macular edema [1]. In the retina, arterioles share the tunica externa with venules where the vessels intersect, so sclerosis in the arterioles may exert pressure on venule walls and narrow the lumina, reducing blood flow and damaging the endothelium through tangential stress; these changes may cause thrombi and, consequently, BRVO [3]. Acute BRVO is characterized by increased capillary and venule pressure, which disrupts the blood-retinal barrier and enables fluid to enter the retina. In macular edema, this fluid collects in the inner to outer retinal plexiform layer [4,5,6].

As changes in retinal blood flow are at the core of BRVO, understanding them is vital to enable the development of effective treatments. Retinal blood flow can be measured noninvasively and in real time with laser speckle flowgraphy (LSFG) [7,8]. LSFG assesses relative flow volume (RFV), which accurately and reliably depicts the volume of blood flowing in retinal vessels and thus reflects vascular resistance [9,10].

Currently, the main treatment for macular edema in BRVO is injection of an anti-vascular endothelial growth factor (anti-VEGF) drug such as ranibizumab [11,12,13]. Intravitreal ranibizumab injection (IRI) was shown to cause constriction of retinal vessels and decrease RFV and blood flow velocity in the occluded and non-occluded regions [14]. In a previous study, we found a significant decrease in venous RFV in the occluded region from baseline to 1 month after IRI [15].

Because IRI may improve symptoms of macular edema in BRVO by decreasing RFV and vessel diameters, data on changes in these variables after anti-VEGF treatment may reveal a new treatment target for BRVO or a method for evaluating its severity. Therefore, we performed a prospective study to assess the effects of IRI on peripheral RFV and diameters of retinal arteries and veins in occluded and non-occluded regions in BRVO with macular edema. Then, we aimed to determine whether RFV and vessel diameter are correlated with cytokine levels in the aqueous humor.

## 2. Materials and Methods

### 2.1. Patients

We enrolled 37 patients with BRVO (37 eyes) who were to be treated by IRI (Lucentis; 0.5 mg in 0.05 mL; Genentech, Inc., South San Francisco, CA, USA) at the Department of Ophthalmology of Tokyo Medical University, Tokyo, Japan, from June 2017 to July 2019. Patients were eligible for IRI if they had macular edema that affected the fovea, with a central macular thickness (CMT) greater than 300 μm and a best-corrected visual acuity (BCVA) below 25/30. CMT represented the distance from the inner limiting membrane to the pigment epithelium and included any serous retinal detachment, if present. Patients were not able to receive IRI if any of the following were present: macular venule occlusion; glaucoma; aphakia; rubeosis iridis; diabetic retinopathy; clinically significant cataract; ocular infection; vitreous hemorrhage; refractive error below—6.0 D; and a history of ocular inflammation or retinal disease besides BRVO, macular laser photocoagulation, or anti-VEGF intravitreal therapy.

The study was approved by the ethics committee of Tokyo Medical University and performed in accordance with the Declaration of Helsinki (IRB No. H-132). All patients gave written informed consent to participate.

### 2.2. Routine Evaluations

A full ophthalmic evaluation, which included decimal BCVA, fluorescein angiography (FA; Digital Retinal Camera CF-1; Canon, Melville, NY, USA), and spectral-domain optical coherence tomography (Spectralis, Heidelberg Engineering, Heidelberg, Germany), was performed before IRI, and BCVA and optical coherence tomography were repeated as a follow-up evaluation 1 month after IRI. CMT was automatically determined by computer software before and 1 month after IRI.

### 2.3. LSFG

Before and 1 month after IRI, RFV and the width of the retinal blood vessels was measured by LSFG (Softcare, Fukutsu, Japan), as described elsewhere [7,8]. Twenty minutes before LSFG, 0.5% tropicamide and 0.5% phenylephrine hydrochloride were used to dilate the pupil.

LSFG is based on the principle that the ocular fundus reflects laser light, creating a speckled pattern, which is then blurred as the reflected light is scattered by red blood cells in retinal vessels. The LSFG recording device includes a fundus camera with an 830-nm diode laser and a charge-coupled device sensor (750 × 360 pixels). The amount of blurring varies depending on the blood flow, and the variation is used to calculate a mean blur rate (MBR, expressed in arbitrary units (AU)) and thus quantify relative blood flow velocity [9,10,16].

LSFG images were acquired for 4 s at a rate of 30 frames per second, and the LSFG device created a composite map of the mean blood flow. The MBR was calculated as the difference between background choroidal blood flow and overall MBR. Then, the MBR values in a direction transverse to the blood vessel were used by the LSFG analyzer software (version 3.1.6) to determine RFV [9,10,15]. RFV was assessed in the main and branch arteries and veins in the occluded and non-occluded regions before and 1 month after IRI (Figure 1).

### 2.4. Hemodynamics

In healthy individuals, RFV shows a bilinear relationship with ocular perfusion pressure (OPP) within a particular range [17]. First, we calculated mean blood pressure (MBP) from systolic blood pressure (SBP) and diastolic blood pressure (DBP) with the formula MBP = DBP + 1/3 × (SBP − DBP). Then, to exclude physiological responses, we calculated OPP from MBP and intraocular pressure (IOP) with the formula OPP = 2/3 × MBP − IOP.

### 2.5. Cytokine Measurements

We measured levels of the cytokines VEGF, interleukin (IL)-6, IL-8, soluble intercellular adhesion molecule (sICAM)-1, monocyte chemoattractant protein 1 (MCP-1), interferon-inducible 10-kDa protein (IP-10), placental growth factor (PlGF), and platelet-derived growth factor (PDGF)-AA [18,19,20,21,22]. To do so, 25-μL aqueous humor samples were first incubated in the dark for 2 h or, for measurements of PlGF and sICAM1, for 16 to 18 h overnight at room temperature. Capture bead kits (Beadlyte; Upstate Biotechnology, Lake Placid, NY, USA) were used in accordance with the manufacturer’s instructions. For each substance, we generated duplicate standard curves from the reference concentrations in the respective kit. In each patient, samples from baseline and 1 month after IRI were analyzed at the same time, together with control samples, with a suspension array system (xMAP; Luminex Corp., Austin, TX, USA) to ensure that between-run differences did not confound results.

The levels of all factors were above the minimum detectable concentration of the respective assay (VEGF, 0.64 pg/mL; IL-6, 0.29 pg/mL; IL-8, 0.14 pg/mL; sICAM-1, 0.03 ng/mL; MCP-1, 1.2 pg/mL; IP-10, 0.55 pg/mL; PlGF, 0.37 pg/mL; and PDGF-AA, 0.64 pg/mL.

### 2.6. Statistical Analysis

For statistical analysis, BCVA was converted to the logarithm of minimal angle of resolution (logMAR) scale. Results are presented as the mean ± SD. Unpaired continuous variables were analyzed by Student’s *t* test, and continuous variables were compared between baseline and 1 month after IRI by a paired *t* test. Relationships between variables were evaluated by Pearson’s correlation analysis. Two-tailed *p* values below 0.05 were considered to be statistically significant. All analyses were performed with SAS System 9.4 software (SAS Institute Inc., Cary, NC, USA).

## 3. Results

The baseline findings in the 37 patients are shown in Table 1. Hypertension (defined as use of antihypertensive drugs or blood pressure > 140/90 mmHg) was present in the majority of patients (see Table 1 for blood pressure values).

Before IRI, the RFV in the main artery and vein correlated significantly with the sum of that in branch vessels 1 and 2 in the non-occluded (Figure 2A,B) and occluded regions (Figure 2C,D). Similarly, after IRI, RFV in the main artery and vein also correlated significantly with that of the respective branch vessels 1 and 2 in the non-occluded (Figure 3A,B) and occluded regions (Figure 3C,D). In non-occluded regions, RFV in the main arteries and veins and branch arteries and veins 1 and 2 did not change significantly from baseline to 1 month after IRI (Figure 4A,B). In occluded regions, RFV in the main artery and vein decreased significantly from baseline to 1 month after IRI (Figure 4C,D), but RFV in branch arteries and veins 1 and 2 did not change significantly (Figure 4C,D).

The correlations between RFV and levels of cytokines in aqueous humor are shown in Table 2. In both the non-occluded and occluded regions, RFV in branch artery 1 was significantly correlated only with the level of sICAM-1. In the occluded region, RFV in the main artery was significantly negatively correlated with the levels of MCP-1 and PDGF-AA; RFV in the main vein was significantly negatively correlated with the levels of MCP-1, PDGF-AA, IL-6, and IL-8; RFV in branch vein 1 was significantly negative correlated with the aqueous levels of PlGF, MCP-1, IL-6, and IL-8; and RFV in branch vein 2 was significantly negative correlated with the aqueous level of PDGF-AA.

Regarding vessel width, the width of the main arteries and veins and branch vessels 1 and 2 did not change significantly from baseline to 1 month after IRI in non-occluded regions (Figure 5A,B) or occluded regions (Figure 5C,D).

The correlations between vessel widths and levels of cytokines in aqueous humor are shown in Table 3. In both the non-occluded and occluded regions, the width of the main artery was significantly correlated only with the level of PDGF-AA. In the occluded region, the width of the main vein was significantly negatively correlated with the levels of MCP-1 and PDGF-AA; the width of branch vein 1 was significantly negatively correlated with the levels of PlGF, PDGF-AA, and IL-8; and the width of branch vein 2 was significantly negatively correlated with the level of PDGF-AA.

One month after IRI, the results showed a significant improvement in BCVA (to 0.15 ± 0.23; *p* < 0.001) and a significant decrease in CMT (to 269 ± 89.9 μm; *p* < 0.001).

## 4. Discussion

In this study in patients with BRVO and macular edema, the RFV in the main arteries and veins of the non-occluded and occluded regions significantly correlated with the summed RFV in the respective branch vessels 1 and 2 both before and after IRI, indicating normal blood flow in the branch vessels before and after anti-VEGF therapy. However, the RFV in the main artery and vein was significantly smaller in the occluded region after anti-VEGF drug therapy. Taken together, these findings show that RFV measurements by LSFG may provide useful information on the progression of retinal ischemia.

The finding that RFV in the main artery and vein decreased significantly from baseline to 1 month after IRI suggests that ranibizumab decreases retinal blood flow in these vessels. One can hypothesize that IRI may also decrease the flow of blood to more peripheral retinal vessels and thus increase the risk of ischemia. This hypothesis is supported by findings of retinal ischemia after anti-VEGF therapy in patients with central retinal vein occlusion [23,24,25]. Ranibizumab appears to decrease blood flow by causing vasoconstriction, which may further increase vascular resistance in the occluded region and decrease blood flow there. VEGF is known to increase blood flow in the retina [26]; the most likely mechanism by which it does so is by increasing the amount of nitric oxide [27]; this stimulates vasodilation [28] and thus increases blood flow and velocity. As an anti-VEGF agent, ranibizumab probably has the opposite effect, i.e., it causes vasoconstriction, which reduces retinal blood flow. Our findings are in line with other studies that found constriction of retinal blood vessels in response to ranibizumab administration [14,29,30]. However, the present study showed no significant decrease in vessel width after IRI and no significant change from baseline to 1 month after IRI in the RFV in branch vessels 1 and 2, suggesting that peripheral blood flow does not decrease after IRI. In a previous study, we found that IRI significantly reduced RFV in veins passing through the optic disc in the occluded region [15]. Thus, findings to date appear to show that IRI affects blood flow in central but not peripheral blood vessels.

In the occluded region, the RFV in the main vein was significantly negatively correlated with levels of MCP-1, PDGF-AA, IL-6, and IL-8, and the RFV in branch vein 1, with levels of PlGF, MCP-1, IL-6, and IL-8. In an earlier study, we showed that venous RFV in the occluded region passing through the optic disc was significantly negatively correlated with levels of MCP-1, IL-8, and IP-10. Together, these findings suggest that, in the retinal veins of the occluded region, MCP-1 and IL-8 may be the most relevant substances affecting blood flow. The chemokine MCP-1 attracts monocytes and macrophages and is upregulated by tangential stress in vitro [31,32]. The movement of monocytes and macrophages into blood vessel walls increases vessel wall permeability, thus leading to or increasing macular edema and decreasing blood flow [6,33,34]. The cytokine IL-8 is produced by vascular endothelial cells, e.g., in response to hypoxia and oxidative stress, and attracts and activates neutrophils and T cells [35,36,37]. In addition, IL-8 also plays a role in leukocyte adhesion to vascular endothelium [38,39]. In an earlier study, we showed a significant correlation between levels of MCP-1 and IL-8 in aqueous humor in BRVO [18]. In an in vivo experiment in rats, retinal vein occlusion increased the rolling and venous wall adhesion of leukocytes, markedly decreasing blood flow [40], so in BRVO with macular edema, this process may be responsible for decreasing RFV [41].

We were surprised that PDGF-AA was significantly negatively correlated with the width of the main vein and the branch veins 1 and 2 in the occluded region. PDGF is a mural cell mitogen, growth factor, and chemoattractant released by endothelial cells that supports pericytes and is thus involved in neovascularization [42,43,44,45,46,47]. Mice with knockout of endothelial PDGF have significantly fewer pericytes and, consequently, vascular defects, showing that PDGF is important for pericyte regulation [42,44,48,49,50,51,52,53]. An in vitro study showed that blocking the recruitment of pericytes recruitment decreases deposition of basement membrane matrix, leading to wider vessels [54]. Overall, these findings suggest that anti-PDGF therapy may increase blood flow by increasing vessel width and that PDGF and VEGF inhibitors such as sunitinib and sorafenib should be explored as promising treatments for BRVO.

In the occluded region, PlGF was significantly negatively correlated with not only RFV but also with the width of branch vein 1. Previously, we reported that PlGF in ocular fluid is significantly elevated in patients with BRVO and macular edema [6,18]. These findings support the already established role of aflibercept (a VEGF and PlGF inhibitor) in treating patients with BRVO and macular edema. Furthermore, in the occluded region, not only the RFV but also the width of the main vein were significantly negatively correlated with the levels of MCP-1 and IL-8. MCP-1 and IL-8 have been reported to be significantly elevated in patients with BRVO and macular edema [6,18]. These findings suggest that the development of inhibitors that target not only the growth factors VEGF, PIGF, and PDGF, but also inflammatory factors such as MCP-1 and IL-8, may improve treatment in the future.

This study has some limitations. The number of cases was relatively small. In cases of severe retinal hemorrhage, LSFG measurement was limited by the difficulty in depicting retinal vessels, so such cases could not be included. Also, samples were collected only at baseline, and cytokines were measured at baseline alone. Therefore, we were unable to evaluate the relationship between changes in cytokines and the changes in RFV and width.

## 5. Conclusions

In summary, this study in patients with BRVO and macular edema showed that before and after IRI, arterial and venous RFV in both the non-occluded and occluded regions correlated significantly with the summed RFV in the respective branch vessels 1 and 2. Furthermore, in occluded regions, RFV in the main artery and vein was significantly lower 1 month after IRI than at baseline. Regarding cytokines, in the occluded region, RFV in the main vein and branch vein 1 was significantly negatively correlated with the aqueous humor levels of various factors. In the occluded region, the width of the main vein and branch veins 1 and 2 was significantly negatively correlated with growth factor PDGF-AA. Thus, this study found that anti-VEGF treatment does not affect peripheral retinal blood in vessel branches but may affect blood flow in the main retinal artery and vein, that cytokines may affect blood flow, and that PDGF may decrease the width of retinal veins.

## Figures and Tables

**Figure 1 medicina-59-01053-f001:**
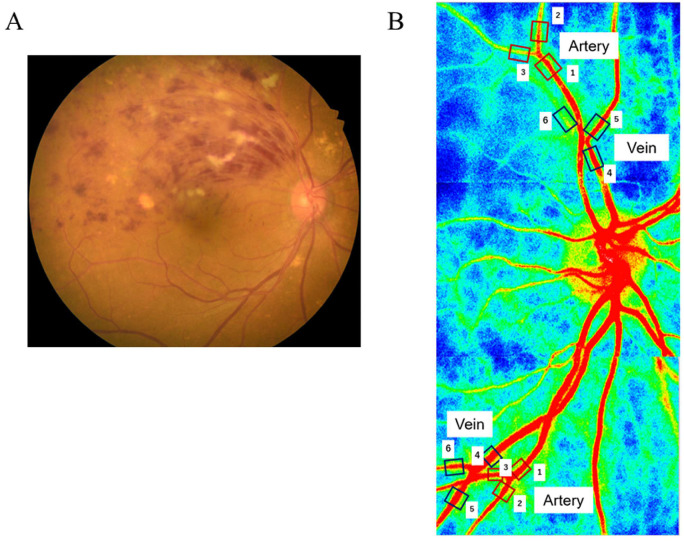
Representative fundus color photograph and representative data on relative flow volume measured by laser speckle flowgraphy (**A**) Fundus color photograph showing branch retinal vein occlusion (BRVO) (top part, occluded region; bottom part, non-occluded region). (**B**) Blood flow in the occluded region: main artery (white square #1) and branch arteries 1 (white square #2) and 2 (white square #3) and main vein (white square #4) and branch veins 1 (white square #5) and 2 (white square #6). Blood flow in the non-occluded region: main artery (white square #1) and branch arteries 1 (white square #2) and 2 (white square #3) and main vein (white square #4) and branch veins 1 (white square #5) and 2 (white square #6).

**Figure 2 medicina-59-01053-f002:**
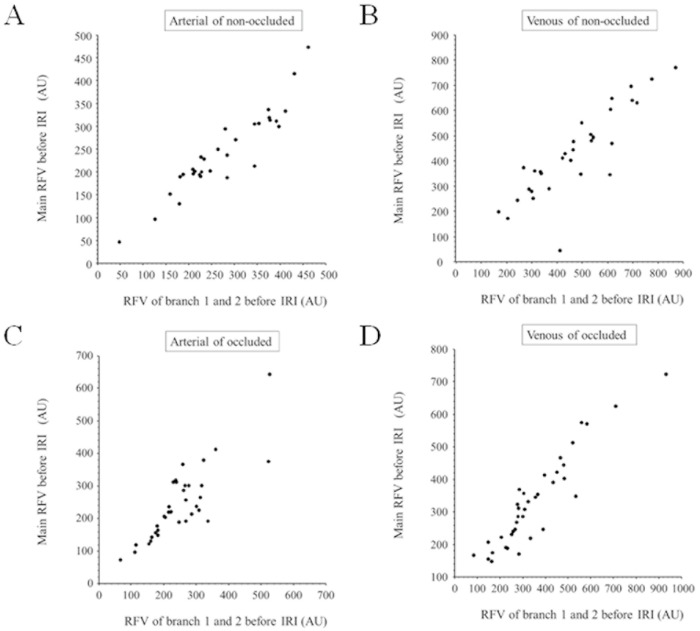
Correlation between relative flow volume in main vessel and branch vessels 1 and 2 before intravitreal ranibizumab injection (**A**) Before intravitreal ranibizumab injection (IRI), relative flow volume (RFV) in the main artery in the non-occluded region correlated significantly with the summed RFV in branch arteries 1 and 2 (r = 0.93, *p* < 0.001). (**B**) Before IRI, RFV in the main vein in the non-occluded region correlated significantly with the summed RFV in branch veins 1 and 2 (r = 0.85, *p* < 0.001). (**C**) Before IRI, RFV in the main artery in the occluded region correlated significantly with the summed RFV in branch arteries 1 and 2 (r = 0.81, *p* < 0.001). (**D**) Before IRI, RFV in the main vein in the occluded region correlated significantly with the summed RFV in branch veins 1 and 2 (r = 0.93, *p* < 0.001).

**Figure 3 medicina-59-01053-f003:**
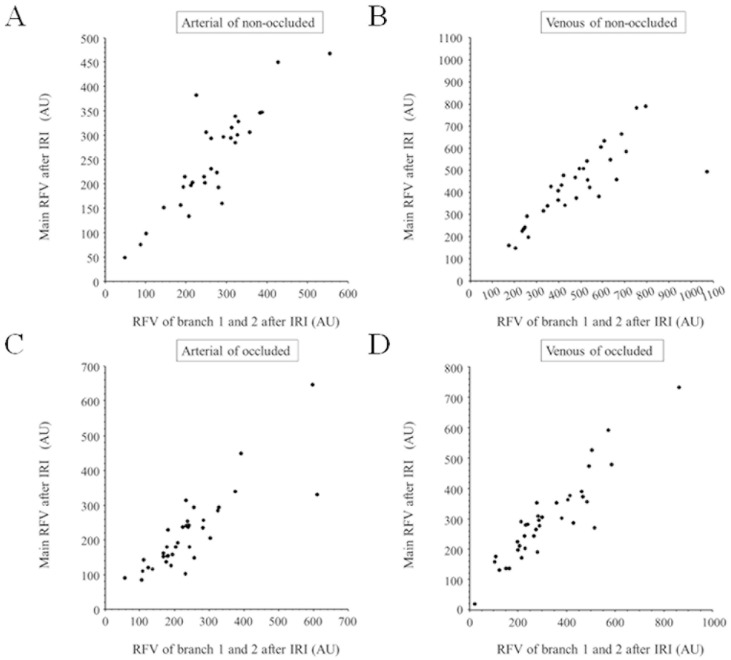
Correlation between relative flow volume in main vessel and branch vessels 1 and 2 after intravitreal ranibizumab injection (**A**) After intravitreal ranibizumab injection (IRI), relative flow volume (RFV) in the main artery in the non-occluded region correlated significantly with the summed RFV in branch arteries 1 and 2 (r = 0.88, *p* < 0.001). (**B**) After IRI, RFV in the main vein in the non-occluded region correlated significantly with the summed RFV in branch veins 1 and 2 (r = 0.80, *p* < 0.001). (**C**) After IRI, RFV in the main artery in the occluded region correlated significantly with the summed RFV in branch arteries 1 and 2 (r = 0.84, *p* < 0.001). (**D**) After IRI, RFV in the main vein in the occluded region correlated significantly with the summed RFV in branch veins 1 and 2 (r = 0.92, *p* < 0.001).

**Figure 4 medicina-59-01053-f004:**
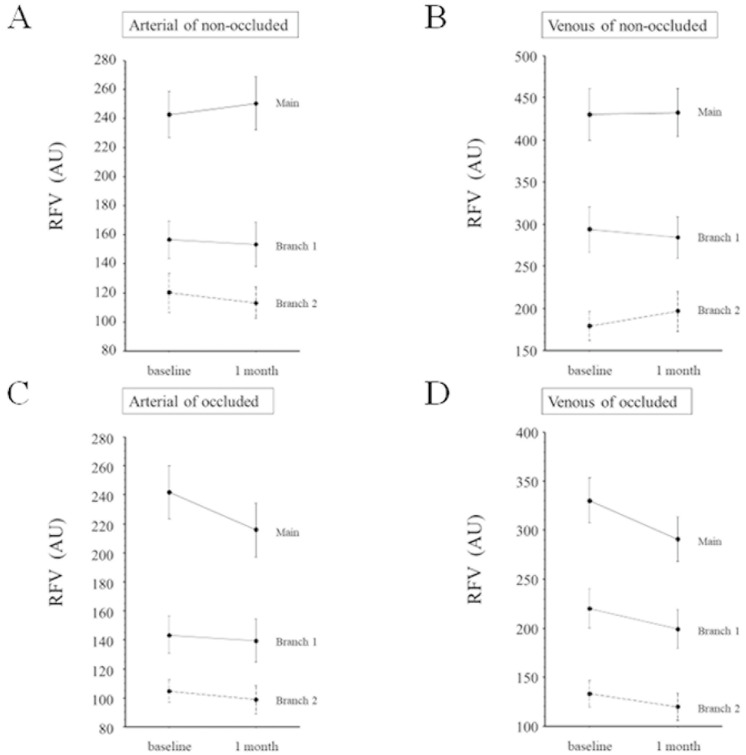
Change in relative flow volume from baseline to 1 month after intravitreal ranibizumab injection (**A**) Relative flow volume (RFV) of the main artery and branch arteries 1 and 2 in the non-occluded region did not significantly change from baseline (main artery, 243 ± 88.3 AU; branch artery 1, 156 ± 72.8 AU; branch artery 2, 120 ± 74.5 AU) to 1 month after intravitreal ranibizumab injection (IRI; main artery, 250 ± 101 AU, *p* = 0.599; branch artery 1, 153 ± 85.3 AU, *p* = 0.679; branch artery 2, 113 ± 60.6 AU, *p* = 0.263). (**B**) RFV in the main vein and branch veins 1 and 2 in the non-occluded region did not significantly change from baseline (main vein, 430 ± 171 AU; branch vein 1, 294 ± 152 AU; branch vein 2, 179 ± 98.7 AU) to 1 month after IRI (main vein, 432 ± 163 AU, *p* = 0.937; branch vein 1, 284 ± 141 AU, *p* = 0.560; branch vein 2, 196 ± 135 AU, *p* = 0.399). (**C**) RFV in the main artery in the occluded region decreased significantly from baseline (242 ± 108 AU) to 1 month after IRI (216 ± 110 AU; *p* = 0.008), but no significant change from baseline to 1 month after IRI was seen in RFV in branch artery 1 (143 ± 76.9 AU vs. 139 ± 89.5 AU, respectively; *p* = 0.553) or 2 (104 ± 46.6 AU vs. 98.6 ± 59.3 AU, respectively; *p* = 0.285). (**D**) RFV in the main vein in the occluded region decreased significantly from baseline to 1 month after IRI (330 ± 140 AU vs. 291 ± 138 AU, respectively; *p* = 0.026), but no significant change from baseline to 1 month after IRI was seen in RFV in branch vein 1 (220 ± 122 AU vs. 199 ± 118 AU, respectively; *p* = 0.144) or 2 (133 ± 81.2 AU vs. 119 ± 84.8 AU, respectively; *p* = 0.095).

**Figure 5 medicina-59-01053-f005:**
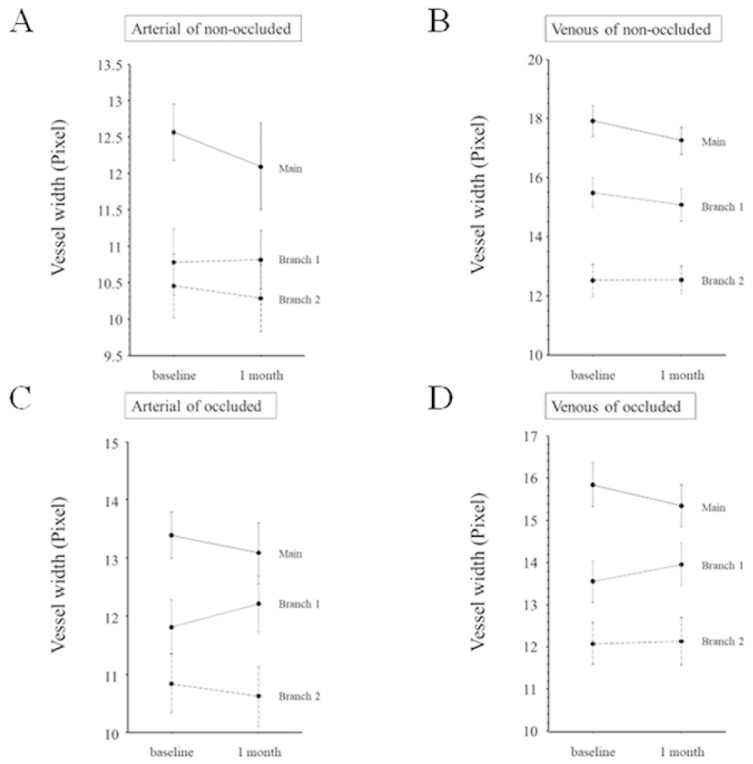
Change in vessel width from baseline to 1 month after intravitreal ranibizumab injection (**A**) The width of the main artery and branch vessels 1 and 2 in the non-occluded region did not change significantly from baseline (main artery, 12.6 ± 2.12 pixel; branch artery 1, 10.7 ± 2.53 pixel; branch artery 2, 10.5 ± 2.46 pixel) to 1 month after intravitreal ranibizumab injection (IRI; main artery, 12.1 ± 3.32 pixel, *p* = 0.405; branch artery 1, 10.8 ± 2.26 pixel, *p* = 0.928; branch artery 2, 10.3 ± 2.54 pixel; *p* = 0.654). (**B**) The width of the main vein and branch veins 1 and 2 in the non-occluded region did not change significantly from baseline (main vein, 17.9 ± 2.86 pixel; branch vein 1, 15.5 ± 2.82 pixel; branch vein 2, 12.5 ± 3.04 pixel) to 1 month after IRI (main vein, 17.2 ± 2.55 pixel, *p* = 0.146; branch vein 1, 15.1 ± 3.04 pixel, *p* = 0.218; branch vein 2, 12.5 ± 2.59 pixel, *p* = 0.946). (**C**) The width of the main artery and branch arteries 1 and 2 in the occluded region did not change significantly from baseline (main artery, 13.4 ± 2.37 pixel; branch artery 1, 11.8 ± 2.76 pixel; branch artery 2, 10.8 ± 2.99 pixel) to 1 month after IRI (main artery, 13.1 ± 3.16 pixel, *p* = 0.419; branch artery 1, 12.2 ± 2.91 pixel, *p* = 0.291; branch artery 2, 10.6 ± 3.09 pixel, *p* = 0.677). (**D**) The width of the main vein and branch veins 1 and 2 in the occluded region did not change significantly from baseline (main vein, 15.8 ± 3.08 pixel; branch vein 1, 13.6 ± 2.97 pixel; branch vein 2, 12.1 ± 2.98 pixel) to 1 month after IRI (main vein, 15.4 ± 3.04 pixel, *p* = 0.0.78; branch vein 1, 14.0 ± 3.03 pixel, *p* = 0.410; branch vein 2, 12.1 ± 3.41 pixel, *p* = 0.863).

**Table 1 medicina-59-01053-t001:** Baseline Clinical Features of BRVO.

Findings	BRVO (*n* = 37)
Age (years)	65.7 ± 8.36 ^‡^
Gender (female/male)	19/18
Duration of macular edema (days)	51.1 ± 44.9 ^‡^
Hypertension	26 (70.3%)
Systolic Blood pressure (mmHg)	141 ± 19
Diastolic Blood pressure (mmHg)	82 ± 13
Hyperlipidemia	16 (43.2%)
Baseline BCVA (logMAR)	0.43 ± 0.29 ^‡^
Baseline CMT (μm)	622 ± 180 ^‡^
MBP (mmHg)	102 ± 14 ^‡^
OPP (mmHg)	55 ± 9.0 ^‡^

BRVO = branch retinal vein occlusion; BCVA = best-corrected visual acuity; CMT = central macular thickness; logMAR = logarithm of the minimum angle of resolution; MBP = mean blood pressure; OPP = ocular perfusion pressure; ^‡^ Mean ± standard deviation (SD).

**Table 2 medicina-59-01053-t002:** Aqueous Humor Levels of Factors/Cytokines and the Relative Flow Volume.

Aqueous Factors/Cytokines	VEGF (pg/mL)	PlGF (pg/mL)	PDGF-AA (pg/mL)	sICAM-1 (pg/mL)	MCP-1 (pg/mL)	IL-6 (pg/mL)	IL-8 (pg/mL)	IP-10 (pg/mL)
Variable	r *p* value	r *p* value	r *p* value	r *p* value	r *p* value	r *p* value	r *p* value	r *p* value
Arterial of the non-occluded region: main	−0.01 0.966	−0.01 0.965	−0.04 0.829	0.17 0.330	−0.26 0.141	−0.03 0.877	−0.05 0.798	−0.08 0.701
Arterial of the non-occluded region: branch 1	0.08 0.644	0.03 0.824	0.11 0.538	0.42 0.011	0.14 0.416	0.09 0.611	0.27 0.124	0.22 0.270
Arterial of the non-occluded region: branch 2	0.03 0.828	−0.10 0.567	−0.01 0.956	−0.08 0.646	−0.32 0.068	−0.12 0.497	−0.18 0.297	−0.22 0.268
Arterial of the occluded region: main	−0.01 0.958	−0.22 0.191	−0.37 0.025	−0.04 0.799	−0.37 0.026	−0.22 0.181	−0.32 0.059	−0.30 0.102
Arterial of the occluded region: branch 1	−0.03 0.860	−0.22 0.181	−0.21 0.210	−0.08 0.627	−0.25 0.132	−0.15 0.375	−0.20 0.226	−0.12 0.552
Arterial of the occluded region: branch 2	−0.02 0.902	−0.09 0.590	−0.24 0.156	0.24 0.152	−0.21 0.206	−0.18 0.290	−0.22 0.178	−0.18 0.296
Venous of the non-occluded region: main	0.10 0.550	0.15 0.369	−0.11 0.541	−0.31 0.065	−0.02 0.882	0.11 0.538	0.13 0.465	−0.14 0.484
Venous of the non-occluded region: branch 1	0.02 0.907	0.14 0.409	0.09 0.593	−0.10 0.552	0.05 0.781	0.15 0.376	0.09 0.594	0.16 0.423
Venous of the non-occluded region: branch 2	0.21 0.204	0.04 0.794	−0.13 0.427	0.15 0.370	−0.06 0.702	−0.12 0.468	0.12 0.466	−0.20 0.320
Venous of the occluded region: main	−0.20 0.224	−0.28 0.096	−0.55 <0.001	−0.08 0.629	−0.44 0.006	−0.38 0.019	−0.48 0.003	−0.23 0.170
Venous of the occluded region: branch 1	−0.14 0.400	−0.35 0.030	0.30 0.064	−0.11 0.495	−0.36 0.028	−0.37 0.023	−0.52 <0.001	−0.20 0.237
Venous of the occluded region: branch 2	−0.28 0.085	−0.05 0.736	−0.43 0.007	−0.07 0.655	−0.16 0.331	−0.19 0.237	−0.12 0.467	−0.03 0.844

VEGF = vascular endothelial growth factor; PlGF = placental growth factor; PDGF = platelet-derived growth factor; sICAM = soluble intercellular adhesion molecule; MCP = monocyte chemotactic protein; IL = interleukin; IP = interferon-inducible 10-kDa protein; r = correlation coefficient. Spearman’s rank-order correlation coefficients were calculated.

**Table 3 medicina-59-01053-t003:** Aqueous Humor Levels of Factors/Cytokines and the Vessel Width.

Aqueous Factors/Cytokines	VEGF (pg/mL)	PlGF (pg/mL)	PDGF-AA (pg/mL)	sICAM-1 (pg/mL)	MCP-1 (pg/mL)	IL-6 (pg/mL)	IL-8 (pg/mL)	IP-10 (pg/mL)
Variable	r *p* value	r *p* value	r *p* value	r *p* value	r *p* value	r *p* value	r *p* value	r *p* value
Arterial of the non-occluded region: main	0.11 0.504	−0.17 0.322	0.35 0.038	0.15 0.376	−0.04 0.799	−0.11 0.501	0.02 0.894	0.03 0.852
Arterial of the non-occluded region: branch 1	0.15 0.369	−0.08 0.641	0.26 0.124	0.23 0.182	0.11 0.531	−0.02 0.890	0.13 0.449	0.15 0.436
Arterial of the non-occluded region: branch 2	0.22 0.198	−0.15 0.372	0.24 0.166	−0.01 0.986	−0.18 0.293	−0.24 0.159	−0.13 0.454	−0.24 0.168
Arterial of the occluded region: main	0.08 0.616	0.05 0.763	−0.03 0.827	−0.10 0.554	−0.11 0.499	0.09 0.581	0.01 0.975	−0.19 0.243
Arterial of the occluded region: branch 1	0.05 0.747	−0.04 0.785	0.17 0.313	−0.03 0.842	−0.10 0.549	0.05 0.740	0.06 0.688	−0.10 0.571
Arterial of the occluded region: branch 2	−0.03 0.824	0.31 0.060	−0.03 0.850	0.08 0.616	0.17 0.304	0.28 0.095	0.17 0.313	0.18 0.308
Venous of the non-occluded region: main	0.12 0.472	0.27 0.118	0.12 0.461	0.08 0.633	0.12 0.457	0.08 0.631	0.27 0.108	0.01 0.941
Venous of the non-occluded region: branch 1	0.12 0.479	0.25 0.137	0.29 0.084	−0.01 0.953	0.10 0.552	0.18 0.297	0.23 0.176	0.18 0.301
Venous of the non-occluded region: branch 2	0.05 0.774	0.11 0.507	−0.15 0.387	0.08 0.637	−0.20 0.233	−0.16 0.345	0.01 0.946	−0.30 0.073
Venous of the occluded region: main	0.04 0.781	−0.01 0.931	−0.56 <0.001	−0.09 0.570	−0.33 0.044	−0.15 0.370	−0.21 0.200	−0.11 0.508
Venous of the occluded region: branch 1	−0.15 0.363	−0.38 0.017	−0.36 0.024	−0.15 0.357	−0.27 0.098	−0.29 0.077	−0.47 0.003	−0.22 0.194
Venous of the occluded region: branch 2	−0.13 0.421	0.05 0.748	−0.33 0.042	−0.09 0.562	−0.15 0.359	−0.14 0.379	−0.06 0.716	−0.01 0.925

VEGF = vascular endothelial growth factor; PlGF = placental growth factor; PDGF = platelet-derived growth factor; sICAM = soluble intercellular adhesion molecule; MCP = monocyte chemotactic protein; IL = interleukin; IP = interferon-inducible 10-kDa protein; r = correlation coefficient. Spearman’s rank-order correlation coefficients were calculated.

## Data Availability

The datasets used and/or analyzed in the present study are available on request from the corresponding author.

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
