# Peer review of "Effects of Intravitreal Ranibizumab Injection on Peripheral Retinal Microcirculation and Cytokines in Branch Retinal Vein Occlusion with Macular Edema"

_medicina, 2023, doi:10.3390/medicina59061053_

Round 1
Reviewer 1 Report
1. In this study on patients with BRVO and macular edema, the authors found that relative flow velocities (RFV) of main artery and vein in the occluded region decreased significantly from before to 1 month after intravitreal ranibizumab injection (IRI). However, there was no significant decrease in vessel width and no significant change in RFV in branch vessels 1 and 2 from baseline to 1 month after IRI. This suggests that IRI affects blood flow in central but not peripheral blood vessels. These findings corroborated with their earlier study findings of reduced RFV in veins passing through the optic disc in the occluded region post IRI.
Though these findings are stated correctly in the results, discussion, and conclusion of this manuscript, however in the abstract conclusions there is an ambiguity in language and that needs to be corrected to reflect the results appropriately.
2. The authors have noted the interesting finding in this study of RFV in the main vein being significantly negatively correlated with PDGF-AA (apart from MCP-1, IL-6 and IL-8) and PDGF-AA being significantly negatively correlated with the width of the main vein and branch veins 1 and 2 in the occluded region. Hence, they speculate that anti-PDGF therapy may increase blood flow by increasing vessel width.
These results brings out a prospective therapeutic target in BRVO patients and the possibility of treatment with PDGF and VEGF inhibitors like sunitinib and sorafenib should be explored.
3. Also, this study found that in the occluded region, RFV in branch vein 1 was significantly negatively correlated with PlGF (apart from MCP-1, IL-6 and IL-8). This supports the already established role of aflibercept (VEGF and PlGF inhibitor) in treating patients with BRVO and macular edema.
In this study on patients with BRVO and macular edema, the authors found that relative flow velocities (RFV) of main artery and vein in the occluded region decreased significantly from before to 1 month after intravitreal ranibizumab injection (IRI). However, there was no significant decrease in vessel width and no significant change in RFV in branch vessels 1 and 2 from baseline to 1 month after IRI. This suggests that IRI affects blood flow in central but not peripheral blood vessels. These findings corroborated with their earlier study findings of reduced RFV in veins passing through the optic disc in the occluded region post IRI.
Though these findings are stated correctly in the results, discussion, and conclusion of this manuscript, however in the abstract conclusions there is an ambiguity in language and that needs to be corrected to reflect the results appropriately.
Author Response
Response to Reviewer 1 Comments
Point 1: In this study on patients with BRVO and macular edema, the authors found that relative flow velocities (RFV) of main artery and vein in the occluded region decreased significantly from before to 1 month after intravitreal ranibizumab injection (IRI). However, there was no significant decrease in vessel width and no significant change in RFV in branch vessels 1 and 2 from baseline to 1 month after IRI. This suggests that IRI affects blood flow in central but not peripheral blood vessels. These findings corroborated with their earlier study findings of reduced RFV in veins passing through the optic disc in the occluded region post IRI.
Though these findings are stated correctly in the results, discussion, and conclusion of this manuscript, however in the abstract conclusions there is an ambiguity in language and that needs to be corrected to reflect the results appropriately.
Response 1: Thank you for your suggestion. As you pointed out, the wording was ambiguous in the abstract conclusions, so we have revised it to be more specific, as follows (page 1, lines 27-32): “Contrary to expectations, the study found that anti-VEGF therapy does not affect RFV in arteries and veins in patients with BRVO and macular edema. Furthermore, retinal blood flow is poor in patients with high MCP-1, IL-6, and IL-8. Finally, high PDGF-AA may result in smaller venous diameters and reduced retinal blood flow.”
Point 2: The authors have noted the interesting finding in this study of RFV in the main vein being significantly negatively correlated with PDGF-AA (apart from MCP-1, IL-6 and IL-8) and PDGF-AA being significantly negatively correlated with the width of the main vein and branch veins 1 and 2 in the occluded region. Hence, they speculate that anti-PDGF therapy may increase blood flow by increasing vessel width.
These results brings out a prospective therapeutic target in BRVO patients and the possibility of treatment with PDGF and VEGF inhibitors like sunitinib and sorafenib should be explored.
Response 2: Thank you for your valuable comments. As suggested, we have added the possibility of treatment with PDGF and VEGF inhibitors such as sunitinib and sorafenib to the Discussion section (page 10, lines 305-307).
Point 3: Also, this study found that in the occluded region, RFV in branch vein 1 was significantly negatively correlated with PlGF (apart from MCP-1, IL-6 and IL-8). This supports the already established role of aflibercept (VEGF and PlGF inhibitor) in treating patients with BRVO and macular edema.
Response 3: Thank you for your valuable comments. As suggested, we have added content on PlGF and aflibercept to the Discussion section (page 10, lines 308-312).

Reviewer 2 Report
It is an interesting and novel subject and well-written paper about the “Intravitreal Ranibizumab Injection on Peripheral Retinal Microcirculation and Cytokines in Branch Retinal Vein Occlusion”. The authors should be explained the limitation of this project and more discussed about the clinical application of this basic research.
Dear Editor
It is an interesting and novel subject and well-written paper about the “Intravitreal Ranibizumab Injection on Peripheral Retinal Microcirculation and Cytokines in Branch Retinal Vein Occlusion”. The authors should be explained the limitation of this project and more discussed about the clinical application of this basic research.
Kind regards
Dr. Mitra Akbari
Author Response
Response to Reviewer 2 Comments
It is an interesting and novel subject and well-written paper about the “Intravitreal Ranibizumab Injection on Peripheral Retinal Microcirculation and Cytokines in Branch Retinal Vein Occlusion”.
Point 1: The authors should be explained the limitation of this project and more discussed about the clinical application of this basic research.
Response 1: Thank you for your positive review of our manuscript. As you suggested, we have added the limitations of this project and the clinical application of this basic research to the Discussion section (page 10, lines 305-307 and 312-323).

Reviewer 3 Report
In the present manuscript, the authors showed that in patients with BRVO and macular edema before and after IRI, the non-occluded and occluded regions correlated significantly with the summed RFV in the respective branch vessels 1 and 2. Moreover, in occluded regions, RFV in the main artery and vein was significantly lower one month after IRI than at baseline. In addition, they also showed that RFV in arteries and veins appears not to be affected by anti-VEGF therapy, even though anti-VEGF therapy may affect retinal blood flow. They also concluded that cytokines may influence retinal blood flow, and PDGF-AA may influence venous diameter.
The conclusions are well supported by the data, and in my opinion, the manuscript is suitable for publication with minor corrections.
1. The data interpretations should be in flow, and it is tough to grasp what the authors want to convey to the reader. Such as….Therefore, we performed a prospective study with the aim of assessing the effects of IRI on peripheral RFV and diameters of retinal arteries and veins in occluded and non-occluded regions in BRVO with macular edema and to determine whether RFV and vessel diameter changes were accompanied by changes in cytokine levels in the aqueous humor…..the authors are summarizing everything in a single sentence. English editing is required.
English editing is required.
Author Response
Response to Reviewer 3 Comments
In the present manuscript, the authors showed that in patients with BRVO and macular edema before and after IRI, the non-occluded and occluded regions correlated significantly with the summed RFV in the respective branch vessels 1 and 2. Moreover, in occluded regions, RFV in the main artery and vein was significantly lower one month after IRI than at baseline. In addition, they also showed that RFV in arteries and veins appears not to be affected by anti-VEGF therapy, even though anti-VEGF therapy may affect retinal blood flow. They also concluded that cytokines may influence retinal blood flow, and PDGF-AA may influence venous diameter.
The conclusions are well supported by the data, and in my opinion, the manuscript is suitable for publication with minor corrections.
Point 1: The data interpretations should be in flow, and it is tough to grasp what the authors want to convey to the reader. Such as….Therefore, we performed a prospective study with the aim of assessing the effects of IRI on peripheral RFV and diameters of retinal arteries and veins in occluded and non-occluded regions in BRVO with macular edema and to determine whether RFV and vessel diameter changes were accompanied by changes in cytokine levels in the aqueous humor…..the authors are summarizing everything in a single sentence. English editing is required.
Response 1: Thank you for your positive review of our manuscript. As you suggested, the purpose of the study in the Introduction was difficult to convey to the reader because it was written in one sentence. We have split that sentence into two parts, and it has been edited by a Native English speaker (page 2, lines 61-66).
